# High-Torque Electric Machines: State of the Art and Comparison

**Maryam Alibeik [1,*] and Euzeli C. dos Santos [2]**

1   Department of Electrical and Computer Engineering, Temple University, 1947 N. 12th Street, Philadelphia, PA 19122, USA

2   Department of Electrical and Computer Engineering, Purdue University-Indianapolis, 723 W. Michigan Street, Indianapolis, IN 46202, USA

*   Correspondence: maryam.alibeik@temple.edu; Tel.: +1-3173705153

**Abstract:** The state of the art of high-torque electric motors has been reviewed in this paper. This paper presents a literature review of high-torque density electric machines based on their airgap classifications, which brings a unique consideration to new design ideas to increase torque density. Electric machines are classified into three main groups based on their airgap configuration, i.e., (1) machines with a constant airgap, (2) machines with a variable airgap, and (3) machines with an eccentric airgap. This paper also presents the modeling of a high-torque airgap-less electric motor based on the concept of eccentric airgap. The torque density of this motor has been compared to motors available in the literature review. Among electrical motors with no permanent-magnet, airgap-less electric motors take the lead in terms of torque density, which is almost five times greater than the next motor, "in-wheel for electric vehicle".

**Keywords:** electric motor; electric machine; torque density; optimization

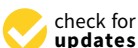



## 1. Introduction

Presently, climate change and its impacts on Earth are some of the main issues that every human should be worried about [1–4]. It is also believed that a big percentage of increased temperature is due to greenhouse gas concentration [5–8]. Based on an article from NASA, $CO_2$ is one of the most important greenhouse gases that can be released through human activities such as burning fossil fuels [9–11]. Scientists have been trying for a long time to reduce the radiation of $CO_2$ worldwide [5,12–15]. Energy consumption is one of the main phenomena that results in the emission of $CO_2$. Recognizing the biggest consumers of energy can help optimize energy consumption. In [16], it is stated that electric machines consume 70% of electrical energy globally. For example, induction machines consume the majority of energy worldwide [17]. One option for optimizing the consumption of energy through electric machines is to increase electric machine efficiency [1].

There is a survey in [18] that focuses mainly on efficiency improvements of induction machines.

Other articles have presented different methods for increasing the efficiency of electric machines and consequently reducing their losses. In [17], the result was shown of reducing temperature [19,20] on increasing the efficiency by forced cooling the coils of an induction motor. This study claims that reducing the coil temperature by 10 °C has an effect of 0.5% efficiency increase. In the same article, it was also shown that internal cooling had more effect on efficiency than external cooling. The use of a unidirectional fan, reformation of the fan blade and fan cover shape, and water or oil cooling are some examples of the cooling methods used in this paper.

Another way to increase the efficiency of electric motors is by using better magnetic materials. As [21] states, conventional soft laminations can be replaced by new materials such as high silicon non-oriented steel, and partially cubic textured steel can increase efficiency by 1.5–3%. Magnetic materials such as NdFeB, which are rare-earth magnets, allow

a very strong magnetic field in a very small volume [22] and a high level of performance. However, these magnets have two main disadvantages that are their expense and the extraction and refinement of rare-earth oxides is a potentially environmentally damaging process, poisoning the farms and villages.

There have been other studies done on the topic of rare-earth magnet-free motors [23–25].

Electric machines can be categorized based on their torque and torque density (torque per motor volume). Achieving high-torque density has always been a target while designing electric motors [26], which has inspired new designs as presented in the technical literature [27]. One of the most important applications for high-torque electric machines is the use of high-torque-density electric machines in an electric vehicle. In other applications, such as industrial and commercial applications, hydraulic motors are selected over electrical motors due to their compact size [28,29]. For example, although the high torque generated by vernier permanent-magnet motors and motors with partitioned rotors are the highest among electric motors, they still do not generate torque density that is comparable with hydraulic motors. Other examples of non-conventional designs of electric machines and control strategies proposed by industry and academia include (1) a dual-rotor structure along with a dual excitation [30], (2) an outer-rotor hybrid excitation [31], (3) new control strategies proposed to enhance mechanical torque [32–35], and (4) injection of third harmonic current [36]. There are also several studies in the literature which focus on improving the torque of electric machines [37–40].

This paper presents a unique classification of electric machines considering their gap characteristics as well as a discussion on high-torque electric motors. Following the introduction, Section 2 will classify high-torque electric machines based on their airgap structure. Section 3 presents the comparison between two electric machines with different excitation methods. Section 4 compares the airgap-less electric motor with a switch-reluctance motor with a different type of excitation.

## 2. Electric Motor Classification Based on Airgap Structure

The existence of the gap in electric machines is unavoidable. The gap is the physical space consisting of air or fluids (such as oils or ferrofluids) which will separate the rotor and the stator of an electric machine. This physical space will allow the rotor to move freely inside or outside the stator. Although the presence of a gap is fundamental in creating the rotational movements, designers often try to minimize this space to maximize the internal flux and consequently electromagnetic torque. Figure 1 shows the classification of electric machines based on their gap structure.

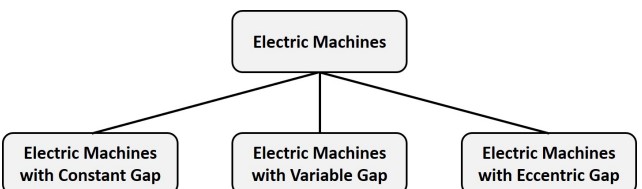

**Figure 1.** Flowchart showing the electric machines classifications.

The gap plays an essential role in defining other parameters of electric machines. For example, the reluctance of the gap in an electric machine is proportional to the length of the gap, as shown in (1). On the other hand, the flux of an electric machine is inversely proportional to the reluctance, as shown in (2), therefore less reluctance will result in more flux.

$$\Re = \frac{g}{\mu_0 A} \tag{1}$$

$$\phi = \frac{Ni}{\Re} \tag{2}$$

Although the least amount of gap is always desirable, in theory it will result in a very large flux, but the length of the gap will never reach zero because of the roughness of the material, and the flux will be limited by the saturation of the magnetic material.

The gap space between stator and rotor can also be filled with ferrofluids instead of air [41–44], as air may cause some limitations in performance due to very low permeability. Additionally, reducing the gap has some negative effects, such as cost, tight tolerances associated with reliability, heat, and dynamic performance which requires enough clearance. Ferrofluids are liquids made of nanoscale ferromagnetic or ferrimagnetic material. These materials have high permeability and consequently will improve the magnetic performance of the machine.

This section presents electric machines' classification according to the structure of their airgap, i.e., (1) constant airgap, (2) variable airgap, and (3) eccentric airgap. In electric machines, the physical geometry of the rotor and stator defines the characteristics of the airgap. In a typical electric machine, the rotor with either constant or variable airgap will only have rotational movement. However, in machines with an eccentric gap, the rotor will have both rotational and translational movements. Both typical and eccentric gap machines will be presented in this work.

### 2.1. Machines with Constant Airgap

Machines with constant airgap will have the same airgap length during one complete rotor cycle either inside or outside the stator.

### 2.1.1. Induction Machines

Induction machines are categorized into two categories—induction machine with squirrel cage rotor, and induction machine with wound rotor. Induction machines with a squirrel cage rotor operate as the voltage is induced in the rotor windings that will produce the rotor current and magnetic field [45–48]. Another type of induction machine is the wound rotor induction machine where the rotor has a three-phase winding inside it. Both machines are identical in terms of electrical characteristics. However, wound rotor induction machines are in disuse due to maintenance issues.

An induction motor with an integrated magnetic gear is also an example of a machine with constant airgap [49]. In this machine, the objective genetic algorithm has been used for torque calculations. Magnetic gears will transmit torque without any contact at all. The magnetic gearing effect is used in [50] to propose a magnetically geared induction machine [51–54]. This machine is shown in Figure 2.

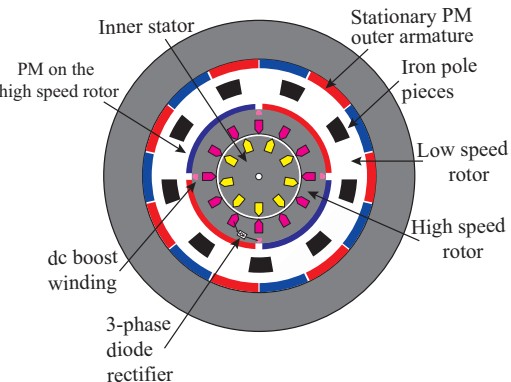

**Figure 2.** Magnetically Geared Induction Machine.

This machine consists of four different armatures of which two are stationary and two are rotating. The innermost armature is the stator of the induction machine. This stator will be excited by a balanced three-phase voltage which will result in creating the magnetic field. This magnetic field will produce an electromagnetic torque in the high-speed rotor. A high-speed rotor consists of magnetic gears that are mounted on its outer surface. These

magnetic gears will then transmit the torque to the low-speed rotor. As shown in Figure 2, the diode rectifier links the machine rotor with DC boost windings to guarantee a 15% increase in torque. Integrating the magnetic gear in induction machines has been done in other studies such as [49,55–57].

### 2.1.2. Synchronous Machines

Another type of electric machine with constant airgap is a synchronous machine. It is worth mentioning that synchronous machines fit in both categories: constant airgap and variable airgap. These machines have a higher torque density among other electric machines [58] which can be employed in low-speed and high-speed applications.

### 2.2. Machines with Variable Airgap

Machines with variable airgap will have different airgap lengths during one complete cycle of the rotor. These machines normally have salient stator, rotor, or both. Figure 3 shows a switch-reluctance motor with variable airgap that has been studied in [59]. As shown, the gap in this machine is changing between the minimum and maximum while the rotor is changing position [59].

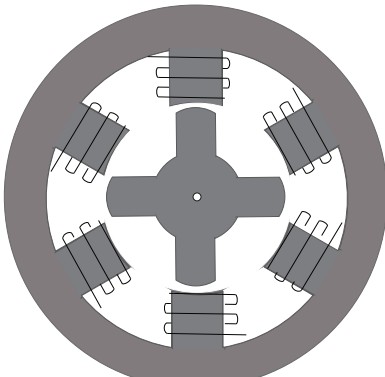

**Figure 3.** Switched-reluctance motor with variable gap.

In these machines, when the rotor tooth is aligned with the tooth of the stator, there will be a smaller airgap than when these teeth are not aligned. In this case, the gap is defined as a variable gap. There exist studies in the literature on electric machines with variable gaps [60,61]. Below are several examples of high-torque electric machines with variable airgaps.

### 2.2.1. Synchronous Machines

As mentioned in the constant airgap section, synchronous machines fit into variable airgap group as well as constant airgap group. In the case that the rotor, stator, or both are salient, the airgap will be variable. In [62] the number of slot/pole combinations of a synchronous machine is studied to find the best combination for low-speed high-torque applications of synchronous machines. Synchronous machines operate either as a motor or a generator. In synchronous generators, the rotor winding is excited by the DC voltage and will induce voltage to the stator. This magnetic field in the rotor winding will rotate the rotor with a constant speed inside the stator. In synchronous motors, in turn, the AC power source will power up the stator which will generate a magnetic field. This magnetic field inside the stator rotates the rotor. Synchronous machines fit in both groups of machines with constant gap and machines with variable gap. Synchronous machines along with induction machines are the most widely used type of AC machines.

### 2.2.2. Permanent-Magnet Machines

Permanent-magnet AC machines are the same type of machines as synchronous machines with the difference that the field windings in synchronous machines are now

replaced with permanent magnets [63]. Permanent-magnet machines have advantages such as fast dynamic performance and high torque or torque per motor volume (torque density), and ease of control relative to alternative machines [64,65]. Interior Permanent-Magnet Synchronous Machines (IPMSM) are desirable for electric vehicle (EV) applications due to their high power and torque density. In [66] the shape of the rotor slots that contain the magnet bars has been optimized to reduce the cogging torque and increase the torque. PM machines conventionally are classified into two different types, i.e., permanent-magnet material on the rotor and permanent-magnet material on the stator. In [67], a permanent-magnet machine is proposed with PM materials on both rotor and the stator. This paper claimed that having the dual excitation would result into a higher torque per volume. Using an extra set of permanent magnets on the ferromagnetic segments of the stator in a magnetic gear was studied in [57]. This study claimed that by adding an extra set of permanent magnets on the stationary part, the torque density would increase by 20%. Figure 4 shows this high-torque density triple permanent-magnet excited the magnetic gear. In this machine, there exists two rotors—the inner rotor at high speed and outer rotor at low speed. The ferromagnetic segment is placed between these rotors and modulates the magnetic field in the airgap between the ferromagnetic segments and the rotors.

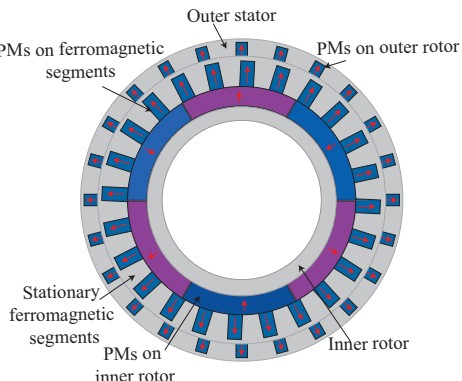

**Figure 4.** Triple PM excited Magnetic Gear.

One of the possibilities to achieve a higher torque is to maximize the radius of the rotor. An outer-rotor claw pole stator has been proposed in [68] in which the torque density is higher than conventional machines due to the radius of the rotor and the claw pole structure of the stator. However, there are also drawbacks for these designs, such as higher stator leakage when the flux is traveling from one pole to the other pole.

### 2.2.3. Interior Permanent-Magnet Synchronous Machines

Interior permanent-magnet synchronous machines [69–73] are among the most popular electric motors used in electric vehicles [23]. These machines contain rare-earth magnetic material which is costly and at the same time has negative impacts on earth. Therefore, one of the demands in the electric vehicle (EV) industry is to decrease the amount of rare-earth magnetic material or electric motors that are rare-earth-free machines. In [23], a switched-reluctance motor has been proposed that does not contain any rare-earth material and is competitive with IPMSM in terms of torque density and efficiency. Although switched-reluctance motors (SRMs) are low cost and simple in design, they have lower torque density compared to other AC machines. This means that this machine produces low torque with respect to its volume. In [59] it has been proposed to employ low-cost magnets on the SRM to increase its torque density and efficiency. It is worth mentioning that a no-rare-earth magnet has been used in this research. The low-cost magnets are in stator yoke to avoid losing the capability of wide speed operation.

### 2.2.4. Synchronous Reluctance Machine

A comparison study between permanent-magnet synchronous machine and synchronous reluctance machine has been done in [74]. In this study, it is evident that in the low power motors, the use of permanent magnets does not change the dimensions of the motor. Also, in cases where weight and size do not matter, the synchronous reluctance machine can be more desirable than the permanent-magnet synchronous machine due to its reliability and low cost. In this paper, it was also mentioned that in motors with high power, having rare-earth magnets is more effective.

### 2.2.5. Permanent-Magnet Flux-Switching Machines

Permanent-magnet flux-switching machines have gained interest in recent decades [75–77]. Although these machines have been proposed in different applications such as wind generation and aerospace, they are known to have high-torque ripples because of the salient stator and rotor. Flux-switching permanent-magnet machines have a stator equipped with both magnets and armature windings. In this case, the rotor will have a robust and simple structure which will make the FSPM machines suitable for high-speed applications [78]. Figure 5 shows the machine that has been studied. This machine operates as a switch-reluctance machine and the electromagnetic performance of this machine is given using FEA.

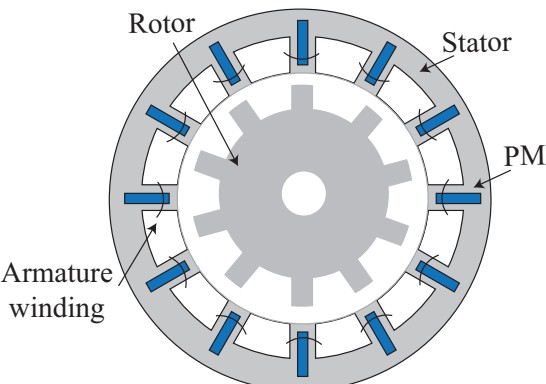

**Figure 5.** Flux-switching permanent-magnet.

In [79] a 36/34-pole nine-phase permanent-magnet flux-switching machine is presented, which will generate higher levels of torque density with lower torque ripples. Changing the rotor configuration to a partitioned rotor in flux-switching PM machines, the stator flux leakage will decrease as well as the use of the PM magnets, which becomes efficient [80]. Due to the high price of rare-earth magnetic material, one of the main goals in developing permanent-magnet machines is to reduce the volume of the magnets, which has been the main purpose in [81].

### 2.2.6. Permanent-Magnet Vernier Machines

One type of permanent-magnet electric machine that has recently gained attention in the literature is the permanent-magnet vernier machine. These machines are normally used in low-speed, high-torque applications [82–85]. Vernier machines are one type of permanent-magnet machine with a difference in the number of stator and rotor poles. It has been claimed in [86] that this type of machine will generate high-torque density compared to regular PM machines due to their special operation principle, which is the magnetic gear effect. Figure 6 shows a typical permanent-magnet vernier machine [87–92].

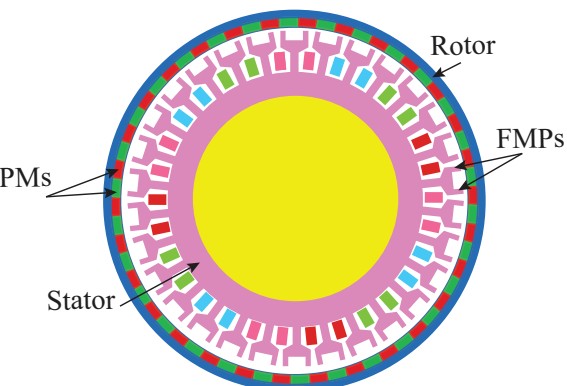

**Figure 6.** Permanent-Magnet Vernier Machine.

In Figure 6, the permanent magnets on the inner and outer rotor produce the inner and outer back-EMFs, respectively. One of the disadvantages of permanent-magnet vernier machines is their higher cost due to the permanent magnets used in stator and rotor poles. Due to this fact, studies presented in the literature have focused on decreasing the cost of the permanent-magnet vernier machine. In [60], a permanent-magnet vernier machine has been proposed which adopts a surface-mounted unipolar rare-earth PM in the rotor, which in turn will reduce the amount of PM used in the machine by half and reduces the flux leakage.

### 2.3. Machines with Eccentric Airgaps

Machines with eccentric airgaps can reach an airgap of almost zero at the point of contact between the stator and the rotor. In these machines, the rotor will have both rotational and transnational movement simultaneously. Despite having these movements, when the rotor touches the stator, the airgap will be almost equal to zero. However, due to the material's roughness, the airgap will never be absolute zero, but has a very small length. It is worth mentioning that even in the simulation of the motors with an eccentric gap, it is not possible to have an airgap of zero as it will result in an infinite force, as shown in (3).

$$f = \frac{ki^2 N}{g^2} \tag{3}$$

Figure 7a shows an electric machine with an eccentric gap that has been studied in [93]. As is evident in Figure 7b, the airgap will reach almost zero at the points of contact between the stator and the rotor [94].

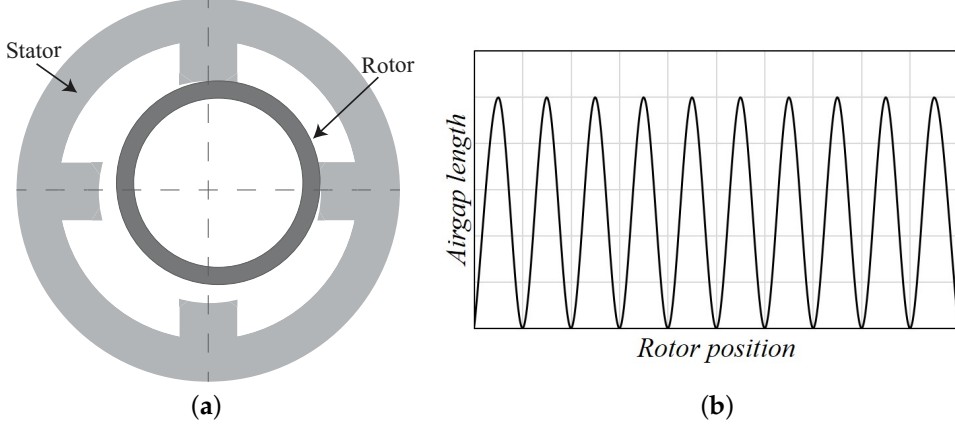

**(a)**                                   **(b)**

**Figure 7.** (**a**) switched reluctance motor with a rolling rotor, and (**b**) eccentric gap vs. position

Reaching a minimal airgap in these machines leads to a higher force and consequently a higher electromagnetic torque. These machines are called rolling-rotor electric machines

in the literature. The authors have studied an airgap-less electric motor with both an external and an internal rotor in [95,96] which will be explained later in this manuscript. Another example of rolling-rotor machines is a rolling-rotor switched-reluctance motor that has been modeled and simulated in [93]. The difference between the modeling and simulation of the rolling-rotor machines in [93,95] is the method of excitation of the stator poles as well as the type of the rotor. In [93], an internal rotor has been used while in [95], an external rotor has been used. An airgap-less electric motor is an eccentric gap type of motor, which is discussed below.

One of the electric motor's main advantages and characteristics is its ability to produce a high torque. However, due to its size, it cannot be used in many applications. In this case, the torque density will be the main subject of interest, i.e., torque per motor volume. Additionally, the two main requirements for electric motors are the high-torque density and high efficiency, as saving energy is the main focus in the world [97].

Based on the type of machine, the stator can have different numbers/types of poles as well as the rotor. In conventional electric machines, the rotor will be connected to a shaft inside the electric machine and will only have rotational movement.

In this section, electric motors have been categorized in terms of their torque density and their torque-producing ability.

Although achieving high torque is important in designing electric motors, presently, torque density has become one of the most important topics among electric motor designers. Since the space issue plays an important role in designing the motor, designers often try to minimize the space and maximize the torque, resulting in a high-torque density (torque per motor volume Nm/L).

## 3. Modeling of the Airgap-Less Electric Motor

The airgap-less electric motor is an eccentric motor designed and modeled by the authors [95,98]. The stator of this motor consists of 18 teeth and nine phases. The bipole configuration has been used to model this motor, i.e., each phase consists of two teeth (north and south). This motor has an external rotor that has both rotational and transnational movement. This family of motors is comparable to hydraulic motors in terms of torque density. Although the name of this motor is the airgap-less electric motor, it does not mean that the gap is zero. Due to the material's roughness, the gap will never reach zero, but it will reach a very small number in the points of contact. As we minimize the gap, the electromagnetic force and torque will increase. This machine will have both rotational and translational movements, so it needs a gearbox to convert this movement to a rotation-only movement. The application of this machine is winches, cranes, or jackhammers. In all three applications, sound and vibration will not be an issue. Additionally, the airgap-less motor has a structure and mechanical configuration similar to a hydraulic motor. Below is a brief review of the analysis that has been done for this machine.

Modeling of this motor starts with obtaining the airgap expression using the geometric approach shown in Figure 8.

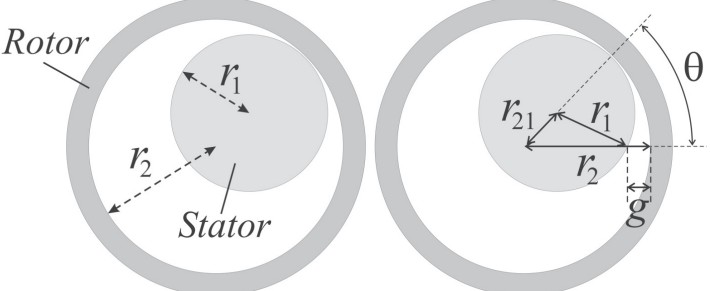

**Figure 8.** Geometric approach used for airgap derivation.

The airgap expression derived using Figure 8 is as shown in (4)

$$g(\theta) = r_2 - r_{21}cos(\theta) - \sqrt{r_1^2 - r_{21}^2 sin^2(\theta)} \tag{4}$$

where $r_{21}$ is the difference between both radii (i.e., $r_{21} = r_2 - r_1$). In this design the radius of the rotor and stator are close to each other, i.e., $r_1 \approx r_2$. Therefore, the gap expression can be simplified as follows:

$$g(\theta) = r_{21}[1 - cos(\theta)] \tag{5}$$

Figure 9 shows the equivalent circuit of the airgap-less electric motor, where $\mathfrak{R}_s$ and $\mathfrak{R}_r$ are the reluctances of the stator and rotor, respectively; $\mathfrak{R}_y$ is the reluctance of the gap $y$ (with $y = 1a, 1b, 2a, 2b, 3a, 3b$); and $\mathfrak{F}_y$ is the magneto-motive-force created at tooth $y$. The reluctances are given by:

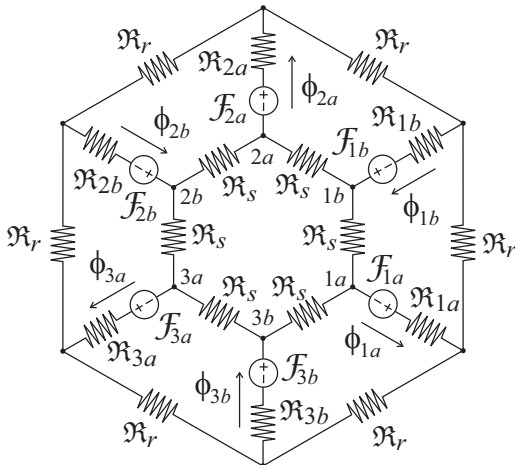

**Figure 9.** Equivalent circuit of the magnetic device.

$$\mathfrak{R}_r = l_r / (\mu_r A_r) \tag{6}$$

$$\mathfrak{R}_s = l_s / (\mu_s A_s) \tag{7}$$

$$\mathfrak{R}_n = g_n(\theta) / (\mu_0 A_s) \tag{8}$$

where $\mu_r$ is the permeability of the rotor, $\mu_s$ is the permeability of the stator, $l_r$ is the length of the flux path in the rotor, $A_r$ is the surface area of the rotor (seen by the flux lines), $l_s$ is the length of the flux path in the stator, $\mu_0$ is the permeability of air, and $n$ is the number of phases for which in this motor it is from 1 to 9.

Please note that in Figure 9, a three-phase version of the actual airgap-less motor is shown due to space limitations. Additionally, for the 18-teeth motor, it is hard to see all the flux paths. Following the derivation of gap and reluctance, the inductance has been calculated using Figure 10 followed by the torque calculation. Figure 11a,b show the current and the electromagnetic torque in the airgap-less motor, respectively.

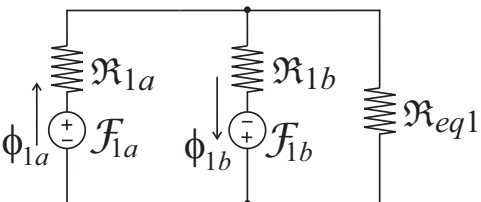

**Figure 10.** Equivalent circuit used for derivation of the inductance.

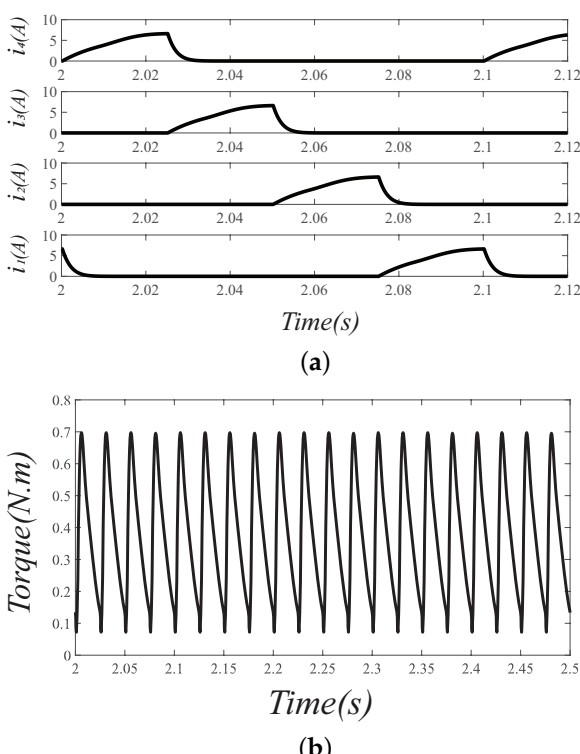

**(a)**

**(b)**

**Figure 11.** Simulation results (nine-bipole machine). (**a**) current, and (**b**) torque.

Figure 12 presents torque densities for six high-torque electric motors available in the literature. The airgap-less electric motor (proposed by the authors) is also among these motors.

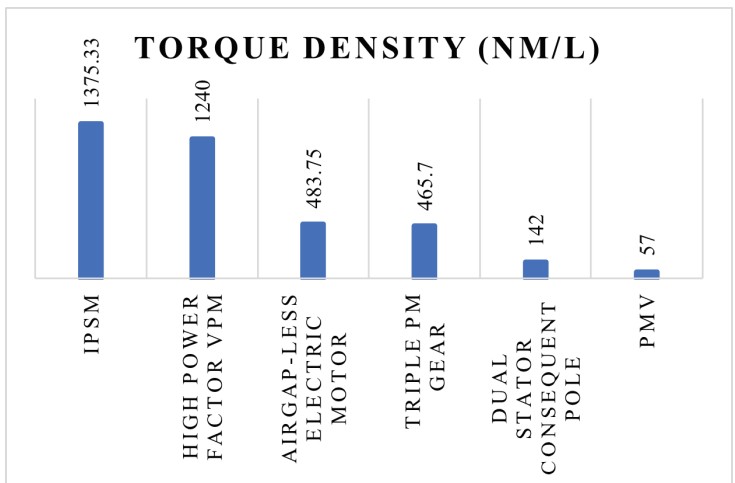

**Figure 12.** Torque density comparison chart for electric motors with permanent magnets.

Except for the airgap-less electric motor, all other motors in this figure consist of permanent magnets, which will result in high torque and, at the same time, will add to the cost of the overall motor. Clearly, in this figure, the airgap-less electric motor does not win in terms of torque density, but as mentioned, there is no permanent magnet in the airgap-less electric motor.

In turn, Figure 13 compares six high-torque electric motors without permanent magnets with the airgap-less electric motor.

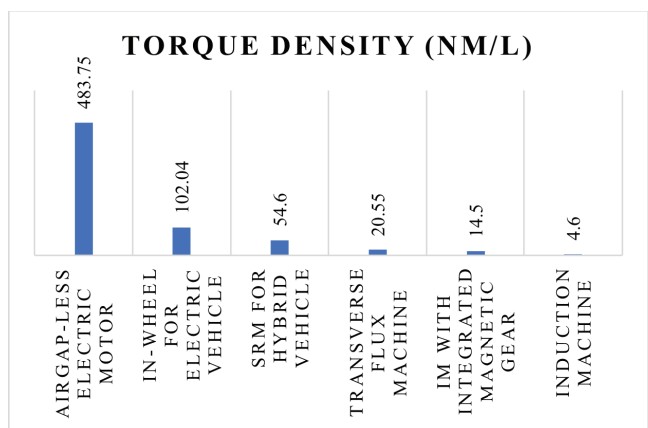

**Figure 13.** Torque Density Comparison Chart for Electric Motors without Permanent Magnet.

In this figure, it is evident that the airgap-less electric motor has the highest torque among other induction and synchronous motors with no permanent magnet.

Table 1 shows the rated power, voltage and speed of these machines.

**Table 1.** Parameters for the machines shown in charts above.

| Machine | Power (W) | Speed (r/min) | Voltage (V) |
|---|---|---|---|
| IPSM | 3728.5 | 1000 | 325 |
| High Power Factor VPM | 2400 | 30 | 27 |
| Dual Stator Consequent Pole | 398 | 300 | 47 |
| Airgap-less Electric Motor | 100 | 0.6 | 20 |
| In-Wheel for Electric Vehicle | 75,000 | 4000 | 900 |
| IM with Integrated MG | 44,000 | 975 | 400 |

## 4. Comparison of Airgap-Less Electric Motors with a Switched- Reluctance Motor

In this section, the airgap-less electric motor will be compared to a switched-reluctance motor studied in [93] in terms of torque-generation capabilities. This switch-reluctance motor has the same movement as the airgap-less electric motor, i.e., rolling rotor. As mentioned in previous sections, the torque density and efficiency of electric machines have recently become the focus in the development of these machines. The airgap-less electric motor fits in the category of rolling-rotor machines. This motor consists of an external rotor and an internal stator.

In the machines with rolling rotors, the airgap between the stator and the rotor varies by the rotation of the rotor. The conventional rotors in the machines that were mentioned previously with variable and constant airgaps only have rotational movements, while in this type of machine (rolling rotor), the rotor has both rotational and translational movement. In these machines, the rotor touches the stator at points of excitation, which will minimize the airgap and consequently maximize the force. In other words, the rotor is attracted to the stator at the points of excitation. The pole excitation of the airgap-less electric motor has been discussed by the authors in detail in [95–98]. It is worth mentioning that the bipole excitation mode has been used in this motor. This means that two teeth will be forming a bipole (north and south) which will create a closed flux path. To have a fair comparison between the airgap-less electric motor and the switch-reluctance motor studied in [93], all the dimensions of both motors and the simulation parameters have been kept the same. The only difference between these two machines is the method of excitation of the stator poles. In the switch-reluctance motor, an alternate form of excitation has been considered. Below are the results of comparison between both machines. Please note that both machines are excited with $V_{dc} = 10$ V. The gaps in both machines are the same since both rotate in the same manner. To avoid confusion in this section, the motor with alternate excitation

mode is called SRM and the motor with bipole excitation mode is called airgap-less. These two machines have different reluctances as they have two different magnetic equivalent circuits. Figure 14a shows the equivalent reluctance of the SRM, while Figure 14b shows the reluctance of the airgap-less electric motor.

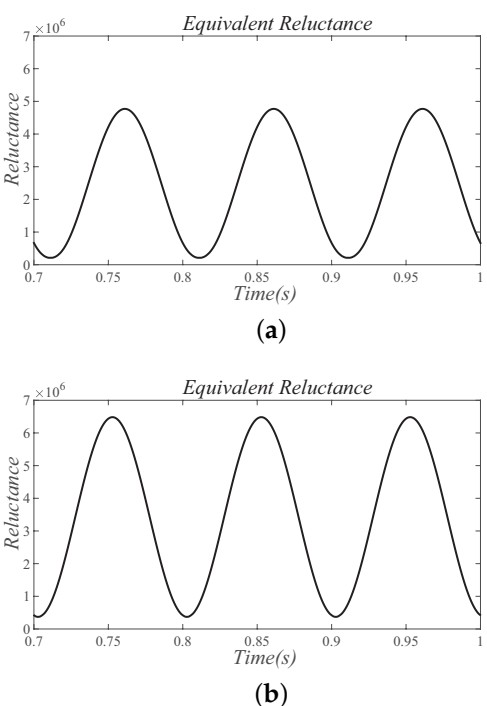

(**a**)

(**b**)

**Figure 14.** (**a**) The machine with alternate excitation mode, (**b**) the machine with bipole excitation mode.

As is evident in Figure 14, the reluctance in the machine with bipole excitation mode is higher and therefore the inductance in this machine is lower, as shown in Figure 15.

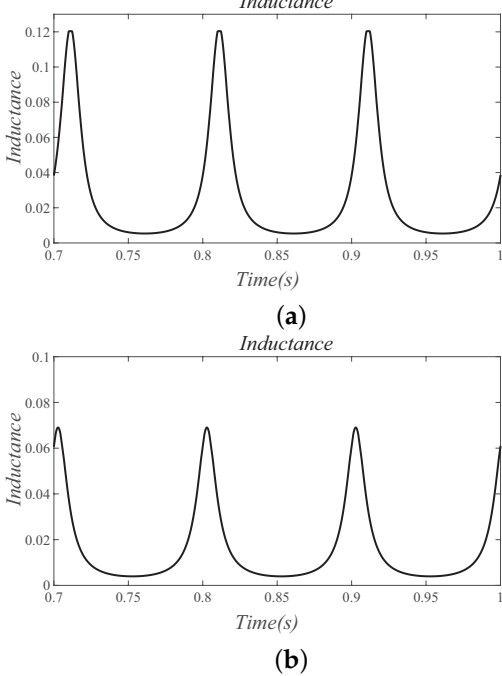

(**a**)

(**b**)

**Figure 15.** (**a**) Inductance of the machine with (**a**) alternate excitation mode and (**b**) bipole excitation mode.

A lower inductance will result in a higher current, as shown in (9).

$$V = L\frac{di}{dt} \tag{9}$$

Figure 16 shows the current in both machines. The co-energy in both machines is then calculated using (10) which leads to the electromagnetic torque as shown in (11).Please note that this is the same equation that has been used in [93] to calculate the electromagnetic torque.

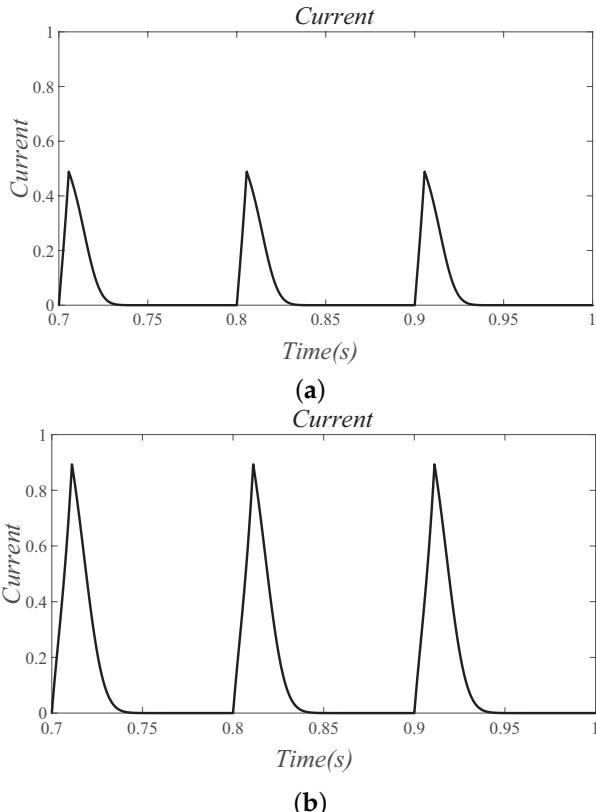

**Figure 16.** (**a**) Current in the machine with (**a**) alternate excitation mode, (**b**) bipole excitation mode.

$$W_c = \frac{1}{2}Li^2 \tag{10}$$

$$T_e = \frac{\partial W_c}{\partial \theta} \tag{11}$$

The electromagnetic torques for both machines are shown in Figure 17, in which it is evident that the electromagnetic torque in the airgap-less electric motor is higher while the torque ripple is lower in the motor with alternate excitation mode, i.e., switched reluctance motor.

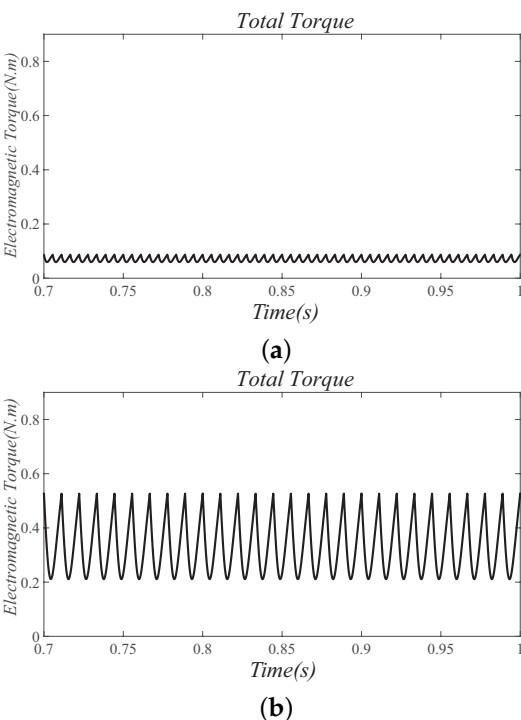

**Figure 17.** (**a**) Torque of the machine with (**a**) alternate excitation mode, (**b**) bipole excitation mode.

## 5. Conclusions

In this manuscript, the state of the art for high-torque electric motors has been presented. The modeling of the airgap-less electric motor previously studied by the authors of this manuscript has been briefly described. This airgap-less electric motor is also compared to both electric motors with permanent magnet and motors with no permanent magnet. The results show that the airgap-less electric motor takes the lead in motors with no permanent magnets in terms of torque density.

**Author Contributions:** Conceptualization, M.A. and E.C.d.S.; methodology, M.A.; software, M.A.; validation, M.A.; formal analysis, M.A.; investigation, M.A.; resources, M.A.; data curation, M.A.; writing—original draft preparation, M.A.; writing—review and editing, M.A.; visualization, M.A.; supervision, E.C.d.S. All authors have read and agreed to the published version of the manuscript.

**Funding:** This research received no external funding.

**Institutional Review Board Statement:** Not applicable.

**Informed Consent Statement:** Not applicable.

**Data Availability Statement:** Not applicable.

**Acknowledgments:** The publication of this article was funded in part by the Temple University Libraries Open Access Publishing Fund.

**Conflicts of Interest:** The authors declare no conflict of interest

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
