# Peer review of "High-Torque Electric Machines: State of the Art and Comparison"

_machines, doi:10.3390/machines10080636_

Round 1

Reviewer 1 Report

Interesting report. Please check the English correctness of your text (i.e. "nowadays" not "Now a days", etc.)

Author Response

Dear Reviewer,

Thank you very much for reviewing the manuscript. I have proofread the manuscript and fixed any spelling and/or grammatical errors.

Best,

Maryam

Reviewer 2 Report

Dear Author

The topic and content of this article have academic research value. The article cites a lot of literature and data, which can comprehensively collect information on high-torque motors, and uses air gaps as classification standards to accurately describe the characteristics of three types of air gap motors, which is innovative to a certain extent. The content of the article is relatively complete, the hierarchical structure is scientific, the main points are prominent, the logical relationship is clear, the language expression is fluent, the format fully meets the requirements of the specification, and no plagiarism is found.

However, when analyzing the difference between air-gap motor and switched reluctance motor, the description is not accurate enough, and it is suggested to modify it slightly.

Author Response

Dear Reviewer,

Thank you very much for reviewing the manuscript.

I have added more explanation in the section " Comparison of Airgap-less Electric Motor with a Switched Reluctance Motor". Please note that in this section I actually used their excitation mode, i.e., alternate excitation mode and compared it to airgap-less motor which used bipole excitation mode. Other than that the number of teeth, and dimensions of both motors are the same.

I have also proofread the manuscript and fixed any spelling and/or grammatical errors.

Best,

Maryam

Reviewer 3 Report

The paper present the literature review of high torque density electric machines based on their airgap classifications.

The research idea is interesting and brings great contributions to the area for presenting comparisons of high torque density engines.

By this reviewer the article can be published.

Author Response

(The authors gave the same response as above.)
